# Nuclear Receptor PXR in Drug-Induced Hypercholesterolemia

**DOI:** 10.3390/cells11030313

**Published:** 2022-01-18

**Authors:** Mikko Karpale, Janne Hukkanen, Jukka Hakkola

**Affiliations:** 1Research Unit of Biomedicine, Biocenter Oulu, Medical Research Center Oulu, University of Oulu and Oulu University Hospital, P.O. Box 5000, FI-90014 Oulu, Finland; mikko.karpale@oulu.fi; 2Research Unit of Internal Medicine, Biocenter Oulu, Medical Research Center Oulu, University of Oulu and Oulu University Hospital, P.O. Box 5000, FI-90014 Oulu, Finland; janne.hukkanen@oulu.fi

**Keywords:** hypercholesterolemia, PXR, SREBP2, PCSK9

## Abstract

Atherosclerosis is a major global health concern. The central modifiable risk factors and causative agents of the disease are high total and low-density lipoprotein (LDL) cholesterol. To reduce morbidity and mortality, a thorough understanding of the factors that influence an individual’s cholesterol status during the decades when the arteria-narrowing arteriosclerotic plaques are forming is critical. Several drugs are known to increase cholesterol levels; however, the mechanisms are poorly understood. Activation of pregnane X receptor (PXR), the major regulator of drug metabolism and molecular mediator of clinically significant drug–drug interactions, has been shown to induce hypercholesterolemia. As a major sensor of the chemical environment, PXR may in part mediate hypercholesterolemic effects of drug treatment. This review compiles the current knowledge of PXR in cholesterol homeostasis and discusses the role of PXR in drug-induced hypercholesterolemia.

## 1. Introduction

Hypercholesterolemia, e.g., elevated LDL cholesterol (LDL-C) and imbalance of LDL-C and high-density lipoprotein cholesterol (HDL-C), is a central causative risk factor of atherosclerosis [1,2]. In fact, the retention of LDL-C and other cholesterol-rich apolipoprotein (Apo) B-containing lipoproteins within the arterial wall represents the key initiating event in atherogenesis [3]. Complications of atherosclerosis, ischemic heart disease and stroke, are among the leading causes of death [4]. In addition to atherosclerosis, excess cholesterol may also be involved in the pathogenesis of other diseases, such as non-alcoholic fatty liver disease and diabetes [5].

Multiple genetic and modifiable factors induce hypercholesterolemia. The most common LDL-C-elevating lifestyle factor is high saturated fat intake, but certain other nutritional factors and obesity also promote hypercholesterolemia [2,3]. Hypercholesterolemia can also be caused by secondary causes, like hypothyroidism [2]. Furthermore, multiple drugs may increase cholesterol. Some drug classes are well-known for their adverse effect on cholesterol, such as antipsychotics and immunosuppressants, but individual drugs from many different classes of drugs have been reported to affect cholesterol levels [2,6]. The mechanisms by which drugs increase cholesterol concentration are still largely unknown.

Nuclear receptor pregnane X receptor (PXR; NR1I2) is a ligand-activated transcription factor that regulates many phase I and II drug-metabolizing enzymes and drug transporters; induction of cytochrome P450 (CYP) 3A4 enzyme is a prime example [7]. Indeed, PXR is an important mediator of the induction type drug–drug interactions [8]. Unlike most other nuclear receptors, PXR accepts a wide array of structurally diverse chemicals as ligands. Many modern synthetic chemicals among environmental contaminants, industrial chemicals, and drugs are well-characterized ligands for PXR. In addition to the classical role in drug metabolism, PXR has been shown to play a role in other biological processes, such as inflammation, cellular proliferation, and glucose and lipid metabolism [9,10]. Due to its pleiotropic functions, PXR is actively studied as a contributing factor in disease pathophysiology and as a novel therapeutic target. Interestingly, PXR activation seems to adversely affect multiple metabolic functions, including lipid metabolism, glucose tolerance, and blood pressure [11]. Thus, it has been hypothesized that PXR activation may partly explain the adverse metabolic effects of environmental chemical exposure and drugs [12,13].

Exposure to environmental chemicals usually consists of a complex mixture of various low-dose exposures with variable amounts and exposure times, making the association of exposure and potential metabolic consequences challenging. In contrast, exposure to drugs is usually well-controlled and both the exposure time and the dose are known. For this reason, the effects of individual drugs on metabolic health are much better known than those of environmental chemicals, and this information could serve as a tool to understand the molecular mechanisms involved and subsequently help to predict the effects of other types of chemicals. This review focuses on the role of PXR in drug-induced hypercholesterolemia. We discuss the evidence for the hypercholesterolemic effect of PXR activation in humans and the mechanistic aspects characterized in mouse models. Finally, we survey the current knowledge on the cholesterol-elevating effect of clinically used drugs and discuss a putative role of PXR in these effects.

## 2. Evidence for Induction of Hypercholesterolemia by PXR Activation in Humans

It has long been known that PXR-activating and CYP enzyme-inducing antiepileptics (carbamazepine, phenytoin, and phenobarbital) [8] elevate cholesterol levels in patients with epilepsy [14,15,16,17]. For example, in one of the earliest prospective studies, phenytoin treatment elevated serum total cholesterol (TC) by 14% at 1-month and 11% at 3-month timepoints in 20 patients with epilepsy [16]. In a subset of 14 patients, 2-year treatment elevated TC by 19%. Similarly, in a prospective study in 36 patients with a more comprehensive lipid panel, 12-month carbamazepine treatment increased mean serum TC by 13%, LDL-C 13%, and HDL-C 14% [14]. In a subset of 19 patients, 5-year prospective measurements also demonstrated elevated levels for TC (by 12%) and HDL-C (26%); the effect on LDL-C was not significant (6%). Most of the cross-sectional studies display cholesterol-elevating effects of enzyme-inducing antiepileptics, while valproic acid, an antiepileptic without enzyme-inducing properties, does not elevate cholesterol [15,17,18,19]. Interestingly, phenobarbital is known to induce human hepatic 3-hydroxy-3-methylglutaryl-CoA reductase (HMGCR) protein, the target of statins, in vivo [20]. However, as enzyme-inducing antiepileptics also activate constitutive androstane receptor (CAR) [8], which is possibly involved in cholesterol metabolism regulation [21], the elevation of cholesterol cannot be solely attributed to PXR. The role of CAR activation alone in cholesterol elevation is difficult to study as there are no selective CAR agonists suitable for human volunteer and patient studies.

Among the drugs that activate PXR, the tuberculosis antibiotic rifampicin is a well-established selective agonist of the receptor [22]. Earlier rifampicin studies with a small number of healthy volunteers (21 days with n = 8, 6 days with n = 10, and 14 days with n = 7, respectively) did not observe an increase in TC [23,24,25], but serum lathosterol-to-cholesterol ratios, a marker of cholesterol synthesis [26], were increased during rifampicin treatment [24]. Furthermore, a trial (n = 12) with cholesterol measurement daily for 30 days reported that a 14-day rifampicin dosing elevated TC by 10% (not statistically significant) with gradual decrease to baseline after discontinuing the dosing [27].

We recently published combined results from two placebo-controlled cross-over trials [28,29] investigating the effect of 600 mg rifampicin for a week on plasma metabolomics, including all lipoprotein fractions and their lipid contents [30]. The study was the largest to date evaluating the effects of PXR activation on plasma metabolomics, with 34 healthy volunteers participating. Rifampicin significantly increased serum TC (by 7%) and LDL-C (12%), with all sizes of LDL particles (large, medium, small) elevated to a similar degree. Esterified (7%) and free cholesterol (7%), as well as intermediate-density lipoprotein (10%), were also elevated by rifampicin dosing. Changes in serum 4β-hydroxycholesterol, a marker of CYP3A4 activity [31] known to be elevated by PXR activation [11], correlated with these changes. The elevations of HDL-C (4%) and total ApoB (5%) by rifampicin were non-significant after correction for multiple testing, while the largest very low-density lipoprotein (VLDL) fractions tended to decrease [30]. Serum concentration of ApoB48, the intestinal form of ApoB, was decreased 14% by rifampicin dosing. During glucose challenge (2 h oral glucose tolerance test), PXR activation by rifampicin resulted in elevated concentrations of very small VLDL particles, remnant cholesterol, and total ApoB [30]. As in a previous study [24], rifampicin elevated serum lathosterol-to-cholesterol ratio [30], and, on the other hand, decreased serum concentrations of cholesterol synthesis precursors citrate and acetate, both suggesting increased cholesterol synthesis.

Another interesting PXR agonist with a strong hypercholesterolemic effect and some human mechanistic evidence on the induction of cholesterol synthesis is mitotane, a steroidogenesis inhibitor used in Cushing’s syndrome and a cytostatic treatment for adrenocortical carcinoma. Although its PXR-activating properties and high risk of drug–drug interactions were revealed only recently [32,33], mitotane has long been recognized as having a remarkable cholesterol-elevating effect. An early study on patients with Cushing’s syndrome demonstrated a 59% increase in serum TC and an 81% increase in LDL-C, with an accompanying 61% increase in ApoB concentration [34]. No significant effects on HDL-C were detected in this small study (n = between 6 and 21). In a recent study in patients with adrenocortical carcinoma (n = 39), mitotane treatment led to 43% increase in TC and 69% increase in LDL-C after 6-month dosing [35]. Additionally, HDL-C was elevated by 14%. Already in 1961, mitotane was suspected to induce cholesterol synthesis as indicated by an increased rate of incorporation of ^l4^C-acetate into plasma cholesterol of patients [36]. Mitotane also elevates plasma levels of mevalonate, a marker of cholesterol synthesis [34].

These findings indicate that (1) drugs with PXR agonism elevate TC and LDL-C and that (2) increased cholesterol synthesis may play a role in drug-induced and PXR-mediated hypercholesterolemia. Genetic studies provide additional evidence, as *PXR* gene polymorphisms associate with plasma LDL-C levels [37].

## 3. Mechanisms of PXR-Induced Hypercholesterolemia

PXR is primarily expressed in the liver and intestine, which are the central tissues for drug metabolism but also those for cholesterol homeostasis. Several intestinal and hepatic mechanisms have been identified that may confer the hypercholesterolemic effect of PXR activation. However, the mechanistic evidence has been mainly gathered utilizing murine models, whose translational value is often limited by substantial differences in murine and human lipoprotein homeostasis. Mice lack cholesterol ester transfer protein (CETP), which leads to high HDL-C and very low LDL-C levels. Nevertheless, the fundamental mechanisms controlling cholesterol synthesis and several other steps in cholesterol homeostasis in humans and mice are similar, and the current evidence indicates important similarities in the mechanisms controlling PXR-mediated elevation of cholesterol in humans and mice. It should be kept in mind that the PXR ligand preference is species-specific and therefore the compound-specific results cannot be directly transferred from mouse or rat experiments to humans.

### 3.1. PXR in Cholesterol Synthesis

In mammals, almost all cells can synthesize cholesterol, but plasma cholesterol is only affected by cholesterol synthesis in the liver due to its central position in lipoprotein metabolism [38,39]. In the liver, as well as in other organs, the regulation of cholesterol synthesis is predominantly determined by the activity of the transcription factor sterol-regulatory element-binding protein 2 (SREBP2) (Figure 1) [40].

Inactive SREBP2 resides in the endoplasmic reticulum (ER) in a complex with SREPB-cleavage activating protein (SCAP) and insulin-induced gene 1 (INSIG1), in which the former functions as a sterol sensor and the latter as a negative regulator of SREBP2 activation [40]. Sterol depletion in the ER incites conformational changes to SCAP, which are required for the translocation of the SREBP2–SCAP complex to the Golgi apparatus. There, SREBP2 is proteolytically cleaved to yield active SREBP2 monomers, which homodimerize, translocate to the nucleus and induce gene expression (Figure 1). In the nucleus, SREBP2 induces genes for cholesterol synthesis, including the rate-controlling enzyme HMGCR. Repletion of sterols in the ER represses SCAP, thus forming a feedback loop to control cholesterol synthesis. In addition to fluctuating ER sterol levels, SREBP2 activity is regulated by several modulators of cellular energy metabolism [41]. For instance, factors affecting INSIG1 expression may affect SREBP2 activation, and SREBP2 is regulated by several post-translational modifications conveyed by hormonal signals and signaling cascades, such as insulin and mammalian target of rapamycin (mTOR). Thus, the need to adjust cholesterol synthesis rates may rise from multiple sources, and cholesterol synthesis presents a highly adaptable biological phenomenon.

Our recently published results indicate that rifampicin treatment increases human plasma cholesterol by activating PXR and hepatic cholesterol synthesis [30]. The same cholesterol synthesis-inducing effect could be repeated also in high-fat diet-fed mice, which were treated with selective murine PXR agonist, pregnenolone-16α-carbonitrile (PCN). This enabled us to study the molecular mechanisms in more detail. In the liver, PXR activation led to nuclear accumulation of active SREBP2 protein, widespread induction of its target genes, and increases in cholesterol and markers of cholesterol synthesis [30].

INSIG1 inhibits SREBP2 activation by retaining the SREBP2–SCAP complex in the ER [42]. PXR has been shown to transcriptionally induce *Insig1* mRNA [43]. With this in mind, the stimulation of SREBP2 activity by PXR was an unexpected finding [30]. We also detected increased *Insig1* mRNA expression in response to PXR activation but were unable to detect increased INSIG1 protein expression [30]. In their study, Roth and colleagues [43] did not report the INSIG1 protein level; furthermore, the study did not evaluate the effect of INSIG1 on the activity of SREBP2 but SREBP1, an SREBP isoform mainly controlling triglyceride synthesis [40]. SREBP1 inhibition by PXR activation is in line with studies describing how PXR induces hepatic steatosis independent of SREBP1 [44,45]. Altogether, these findings suggest that PXR may affect INSIG1 translation or protein degradation and selectively affect SREBP1 and SREBP2 activities. In addition to INSIG1, PXR has been shown to suppress fibroblast growth factor 21 (FGF21), a negative regulator of SREBP2 expression [46,47], which may play an additional role in PXR-stimulated SREBP2 activity, although *Srebp2* mRNA expression remained unaffected in our experiments.

Cholesterol is synthesized from acetyl-CoA, which undergoes multiple enzymatic reactions to form squalene, subsequently converted to (S)-2,3-epoxysqualene by squalene epoxidase (SQLE) and further to lanosterol, which is converted to cholesterol either in the Bloch or in the Kandutsch–Russell pathway [48,49]. Recently, an antiretroviral efavirenz was shown to induce cholesterol synthesis in mice by activating PXR and inducing hepatic *Sqle*, a novel PXR target gene [50]. Importantly, the effect of efavirenz on cholesterol was abolished in mice lacking hepatic PXR. In our study, PCN treatment induced hepatic *Sqle* among other genes of cholesterol synthesis [30]. Furthermore, all PCN-treated mice had lower liver squalene levels than controls, which possibly indicated faster squalene metabolism and increased SQLE activity. Interestingly, besides HMGCR, SQLE is another rate-limiting enzyme of cholesterol synthesis and cholesterol constitutes its inhibitory feedback signal, in addition to downregulation of proteolytic processing of SREBP2, also by directly inhibiting SQLE activity [51,52]. Besides the direct PXR mediated regulation [50], distortion of normal inhibitory regulation of SREBP2 by PXR activation may in part account for the induction of *Sqle* gene expression [30].

We showed that increased cholesterol synthesis due to PXR activation was caused by the induction of the Kandutsch–Russell pathway, as evidenced by increased plasma and hepatic markers of the pathway, lathosterol and zymostenol, and induced DHCR24, an enzyme that directs cholesterol synthesis to the Kandutsch–Russell pathway [30]. PXR did not affect desmosterol, a marker of the Bloch pathway and a negative regulator of SREBP2 [53], meaning that by inducing the Kandutsch–Russell pathway instead of the Bloch pathway PXR may evade normal SREBP2 downregulation by desmosterol (Figure 1). Overall, these results are in line with the previous reports regarding PXR and cholesterol synthesis and, most importantly, with the human findings, and suggest that PXR activation stimulates hepatic SREBP2 activity with harmful effects on circulating atherogenic lipids. 

### 3.2. PCSK9 Induction by PXR Activation

Identification of proprotein convertase subtilisin kexin-type 9 (PCSK9) and development of PCSK9 inhibitors for drug therapy has remarkably improved the understanding of plasma cholesterol regulation [54,55]. PCSK9 is secreted from the liver to the circulation, where it functions to induce the degradation of hepatic LDL receptors. This results in decreased LDL clearance and increased circulating LDL. Inhibition of plasma PCSK9 by antibodies has proven to be an efficient therapeutic strategy to lower plasma LDL [56]. 

In the liver, the expression of *Pcsk9* is regulated by SREBP2 [57]. As PXR activation was shown to stimulate SREBP2 activity, this raised the hypothesis that PXR could also induce *Pcsk9* expression. Indeed, this was found to be the case in both mice and humans [30]. LDL receptor is another SREBP2 target gene and could counteract the negative effect of PCSK9. Expectedly, the LDL receptor mRNA was also induced by PXR activation in mouse liver; however, the response was very minor compared with the *Pcsk9* response [30].

Statins stimulate SREBP2 activity and *Pcsk9* expression as a consequence of cholesterol synthesis inhibition and lower sterol content in the hepatocyte ER. Interestingly, lipophilic statins (atorvastatin, simvastatin) have been shown to increase PCSK9 more efficiently than hydrophilic statins (rosuvastatin, pravastatin) [58]. Atorvastatin and simvastatin are also PXR ligands, and it could be speculated that, in addition to the decreased cellular sterol content, the PXR agonism plays an additional role in the PCSK9 increase by these statins.

The current list of drugs that induce PCSK9 include statins, fibrates, mTOR inhibitors, nilotinib, and rifampicin [30,59,60,61,62]. The mechanisms are probably diverse but given the recently discovered role of PXR in *Pcsk9* regulation, further research on the effects of PXR agonists on *Pcsk9* expression is warranted. Of the currently known PCSK9-inducing drugs, several (atorvastatin, simvastatin, rifampicin, and nilotinib) are PXR ligands, although they involve also other cellular mechanisms.

### 3.3. PXR in Intestinal Cholesterol Absorption

Intestinal absorption of biliary and dietary cholesterol is an important contributor to hypercholesterolemia, and ezetimibe or food additives (plant stanols or sterols) that limit intestinal cholesterol absorption decrease circulating cholesterol [63,64]. Interestingly, Niemann-Pick C1-like 1 (NPC1L1), an intestinal cholesterol transporter protein and the molecular target of ezetimibe, has been shown to be directly regulated by PXR [65]. Tributyl citrate, a common plasticizer, and quetiapine, an atypical antipsychotic with known adverse effects on lipid metabolism, have been shown to induce hypercholesterolemia in mice by activating intestinal PXR, which is associated with increased *Npc1l1* expression [50,65]. It is noteworthy that in mice NPC1L1 is expressed only in the intestine, whereas in humans NPC1L1 is also strongly expressed in the liver, which may affect the translational value of these findings [66]. Rifaximin is a gut-specific human PXR activator, which does not increase serum cholesterol in PXR-humanized mice, although it induces intestinal triglyceride absorption [67,68]. However, rifaximin induced hepatic cytochrome P450 7A1 (CYP7A1) enzyme, the gatekeeper of bile acid synthesis, which suggests that increased bile acid synthesis may have protected the mice from hypercholesterolemia.

### 3.4. PXR in Bile Acid Synthesis and Cholesterol Metabolism

Synthesis of bile acids from cholesterol and biliary secretion of bile acids and cholesterol present a major route of excess cholesterol disposal. Increasing bile acid disposal by inhibiting bile acid absorption from the intestine to the circulation by bile acid sequestrants, such as cholestyramine and colesevelam, lowers plasma cholesterol, highlighting the importance of hepatic cholesterol disposal as a regulatory mechanism of circulating cholesterol.

PXR activation has been shown to have a complex role in endogenous bile acid homeostasis. Activation of PXR by some bile acids is critical to accelerate the detoxification of otherwise hepatotoxic bile acids [69,70,71]. Bile acids are detoxified mainly by known PXR target genes; CYP3A enzymes, bile acid conjugation enzymes SULT2A1 and UGTs, and bile acid transporter MRP2 [72,73].

Several studies have reported that PXR activation represses CYP7A1, the rate-limiting enzyme of bile acid synthesis, thus forming a negative feedback loop to regulate bile acid homeostasis [72,74,75,76]. Although PXR activation represses CYP7A1 in several animal and cell models, rifampicin dosing does not seem to affect CYP7A1 expression in humans [77] and even increases bile acid synthesis, as evidenced by increased serum levels of 7α-hydroxy-4-cholestene-3-one, a marker of bile acid synthesis [24,77]. 

## 4. PXR in HDL Homeostasis

HDL is a central factor in reverse cholesterol transport, a mechanism by which tissues and cells get rid of excess cholesterol; HDL functions to transport cholesterol from extrahepatic tissues to the liver and intestine [78]. Although PXR activation appears to have mainly harmful effect on cholesterol homeostasis, there is some evidence that PXR activation could increase HDL-C. For instance, some drugs with PXR-activating properties, such as carbamazepine and phenytoin, appear to increase HDL-C [14,79,80,81]. Furthermore, we have reported that rifampicin induces CYP3A4-mediated formation of 4β-hydroxycholesterol (4βHC) and, as a liver X receptor (LXR) agonist, 4βHC in turn stimulates cholesterol efflux transporters in macrophages, possibly promoting HDL-C-mediated reverse cholesterol transport [82]. Thus, elevation of circulating 4βHC could partly explain the HDL-C increasing effect of PXR-activating drugs (PXR–4βHC–LXR circuit). For more information on this topic, please see our recent review [11].

## 5. Drug-Induced Hypercholesterolemia

Increase in circulating cholesterol during drug treatment is not uncommon, as several antihypertensives, antihyperglycemics, antipsychotics, antiretrovirals, and immunosuppressants may have an unfavorable effect on cholesterol. Many of these drugs are used long-term, increasing the potential for cardiovascular risk. Drug-induced hypercholesterolemia may be directly caused by the drug or secondarily caused by other effects of the drug, such as weight gain. For example, some antipsychotics, antidepressants, anticonvulsants, and hormones may increase weight and negatively affect cholesterol status [83].

To obtain a systematic overview on drug-induced hypercholesterolemia, we identified clinically used drugs that increase TC or LDL-C in the commercially available Drug Laboratory Effects database (Multirec Ltd., Turku, Finland) (Table 1). To increase the clinical significance of the table, we removed drugs that have been withdrawn from the market or were reported to have both lowering and increasing effects on cholesterol. Furthermore, we added bexarotene, mitotane, and rifampicin to the table, as they have reported effects on cholesterol but were not included in the original database [22,30,35].

Table 1 consists of 106 drugs, of which 48 drugs increase both LDL-C and TC, 54 drugs increase LDL-C, and 100 drugs increase TC. Among the drugs that increase LDL-C, the largest drug classes are immunosuppressants with 15 drugs (28%), antineoplastics with 10 drugs (19%), antipsychotics with eight drugs (15%), and antiretrovirals with six drugs (11%). Together these four classes comprise 72% of the drugs listed.

A similar pattern is seen in the 100 drugs that increase TC (Table 1): 62% of the drugs that increase cholesterol are either antineoplastic (22 drugs; 22%), immunosuppressants (19 drugs; 19%), antiretrovirals (12 drugs; 12%), or antipsychotics (9 drugs; 9%). In addition, the list of drugs that increase TC include several commonly used glucocorticoids, serotonin reuptake inhibitors, and non-steroidal anti-inflammatory drugs. Naturally, the lists of drugs increasing TC and LDL-C are heavily overlapping. It should be noted that sometimes the original publications used as the source for the database did not report all the relevant cholesterol values. Furthermore, in many cases there exist conflicting data on the effect of drugs on plasma cholesterol levels. Finally, among the drugs that have been reported to increase TC and/or LDL-C, the magnitude of effect and the clinical significance are very variable.

Some drug classes are clearly overrepresented in the list of cholesterol-elevating drugs and their effects may seem like a class effect of certain drugs. However, there are still significant differences between the drugs within the therapeutic groups. For instance, a recent meta-analysis described marked differences between atypical antipsychotics in the metabolic side effects, including TC and LDL-C, with olanzapine and clozapine displaying the worst profiles [130]. Furthermore, changing from quetiapine, risperidone, or olanzapine to aripiprazole has been reported to be beneficial for cholesterol status and 10-year cardiovascular disease risk [131]. Notably, these antipsychotics have also significant differences in their target receptor profiles that could potentially play a role in their varying effects on metabolic parameters. 

The mechanistic understanding of the molecular mediators of drug-induced dyslipidemia is elusive and the data to date are limited by the lack of sophisticated human studies. The hypercholesterolemic effects of several antipsychotics have been linked to increased activation of SREBP2 due to inhibition of INSIG-2 in murine models [132]. Furthermore, INSIG-2 gene polymorphisms in schizophrenia patients are associated with weight gain and prevalence of metabolic syndrome [133].

Inflammation and drugs that modulate inflammation, non-steroidal anti-inflammatory drugs (NSAIDS), glucocorticoids, and immunosuppressants, are known regulators of cholesterol levels. Acute infections tend to decrease LDL cholesterol, and the more severe the infection, the more pronounced the effects on cholesterol [134,135]. The same seems to apply to chronic inflammatory diseases, as a decrease in HDL and LDL cholesterol is a common finding in cases of rheumatoid arthritis and the reductions in HDL and LDL correlate with disease severity in ankylosing spondylitis [136,137]. As inflammation attenuates during anti-inflammatory drug treatment, cholesterol levels return to normal, which may explain why many anti-inflammatory drugs seem to increase cholesterol [138,139]. However, some drugs used to treat inflammatory diseases may have more specific effects on lipid metabolism. 

Interestingly, immunosuppressants used in organ transplant patients, cyclosporin (calcineurin inhibitor) and especially the mTOR inhibitors sirolimus, tacrolimus, and everolimus, increase cholesterol [140,141]. Immunosuppressant-mediated dyslipidemia is common and occurs in 60% of organ transplant patients [142]. The mechanisms by which immunosuppressants induce dyslipidemia are not well understood, but they seem to involve LDL metabolism and PCSK9 [143,144]. mTOR is a central regulator of lipid metabolism and SREBP, which already hints at mechanisms of mTOR inhibitor-induced dyslipidemia [145]. 

Antiretroviral therapies have improved the life expectancy of human immunodeficiency virus (HIV)-infected patients. This success has brought up HIV-associated risks for other comorbidities, including increased risk for cardiovascular diseases [146,147]. One significant contributing factor is the adverse effect of antiretrovirals on lipid metabolism [146]. Protease inhibitors, non-nucleoside reverse transcriptase inhibitors, and integrase inhibitors may all increase TC or LDL-C [148]. Most of the approved protease inhibitors and about a quarter of the approved reverse transcriptase inhibitors are included in the list of drugs that increase cholesterol (Table 1). Mechanistically, protease inhibitors have been shown to stimulate SREBP activity in the liver [148]. 

## 6. Identification of PXR Ligands among the Drugs Inducing Hypercholesterolemia

To gain a systematic overview of the potential role of PXR in drug-induced hypercholesterolemia, we identified in the literature the PXR-activating drugs among the drugs that increase TC or LDL-C (Table 1). 

Only studies that provided direct evidence of PXR binding were considered. For instance, induction of classical PXR target genes, CYP3A and CYP2C, was not considered a proof of PXR agonism, as the expression of these genes is also regulated by other nuclear receptors. PXR activation is often evaluated based on CYP induction and consequent drug–drug interactions. However, this may not always reveal actual PXR activation status, as some inducers are also CYP inhibitors [149]. Indeed, several drugs that are PXR agonists also inhibit CYP3A4 activity. In these cases, the other effects of PXR activation could still be significant. On the other hand, the results of reporter gene assays or other in vitro methods may overestimate the receptor activation, especially if performed only with high ligand concentrations. 

Among the 106 drugs that increase either LDL-C or TC, 23 (21.7%) were identified as PXR activators, and three, brigatinib, apalutamide, and ruxolitinib, as likely activators. Thus, altogether, of the 106 drugs that increase cholesterol levels, 26 (24.5%) are most likely to have PXR-activating property. Among the drugs that increase LDL cholesterol but have not been identified as PXR agonists, clozapine, and nevirapine have been shown to have some inducing effect on CYP3A4, but only nevirapine is a clinically significant CYP3A4 inducer. Among the drugs that increase TC, 10 drugs have not been identified as PXR agonists but induce CYP3A4.

PXR and its sister receptor constitutive androstane receptor (CAR) both regulate drug metabolism but also glucose and lipid metabolism and also share some ligands. However, while PXR activation is detrimental to metabolic performance, CAR activation has been shown to be beneficial. Of importance, CAR activation has been shown to lower cholesterol by increasing fecal cholesterol disposal and by suppressing cholesterol synthesis [21,150]. 

## 7. Conclusions

A large number of drugs have been reported to display adverse effects on plasma cholesterol level. For many, the effect size is rather small, but there is still a significant number of drugs for which the effect is considered clinically significant. In long-term treatment, the unfavorable effect of drug therapy on cholesterol may predispose to atherosclerosis, especially if combined with other risk factors. 

Recent studies have indicated that activation of nuclear receptor PXR increases plasma cholesterol and several potential mechanisms have been reported, including increased hepatic cholesterol synthesis, induction of PCSK9, increased intestinal absorption, and decrease of bile acid synthesis. In humans, there is currently evidence for the first two of these mechanisms. So far, rifampicin, efavirenz, and quetiapine have been reported to induce hypercholesterolemia through PXR-mediated mechanisms, either in humans or mice [30,50,96]. Among the known cholesterol-increasing drugs, about a quarter are certain or possible PXR ligands. Thus, PXR activation is likely to at least partly mediate their hypercholesterolemic effects but verifying the role of PXR requires further studies in each case in the future.

## Figures and Tables

**Figure 1 cells-11-00313-f001:**
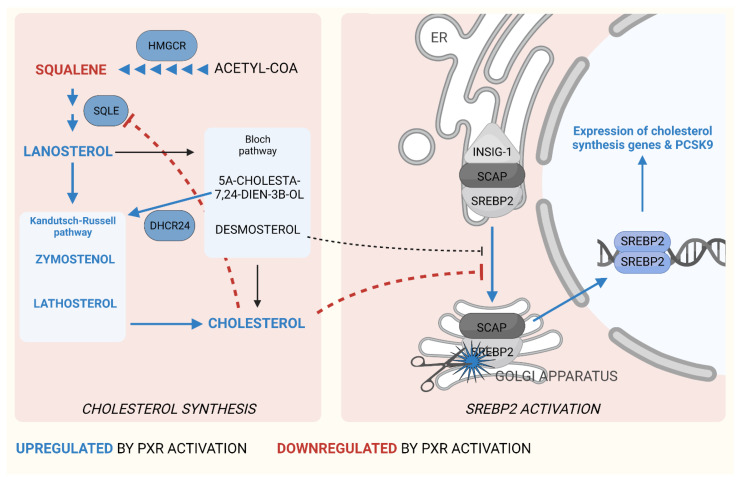
**Activation of PXR stimulates cholesterol synthesis and induces *Pcsk9* in the liver.** The processes and molecules upregulated and downregulated by PXR activation have been indicated with blue and red coloring in the figure, respectively. PXR activation leads to nuclear accumulation of SREBP2, which consequently induces cholesterogenic genes, including the rate-limiting enzyme HMGCR, and thus induces cholesterol synthesis. Furthermore, PXR activation directly induces *Sqle* to increase the rate of squalene epoxidation. Induction of DHCR24 enzyme targets cholesterol synthesis flux to the Kandutsch–Russell pathway instead of the Bloch pathway. PXR activation appears to bypass the usual negative feedback mechanism controlling cholesterol synthesis, including inhibition of SREBP2–SCAP complex translocation from the ER to the Golgi apparatus by high cellular cholesterol levels and desmosterol, as well as the inhibitory effect of cholesterol on SQLE activity. Accumulation of plasma LDL-C level is potentiated by induction of the hepatic *Pcsk9* gene and a consequent increase in circulating PCSK9 levels.

**Table 1 cells-11-00313-t001:** Cholesterol-increasing drugs and their potential to activate PXR and induce CYP3A4.

Drug Class	Drug	Mechanism	Increases	PXR Agonist	CYP3A4 Inducer	Clinically Relevant CYP3A4 Inducer
Androgen	Methyl testosterone	Androgen receptor activation	LDL			
Antiarrhythmic	Amiodarone	Blocking of voltage gated K^+^ and Ca^2+^ channels	LDL			
Antibiotic	Rifampicin	Bacterial RNA synthesis inhibition	CHOL, LDL	Yes [22,84]	Yes [22]	Yes [22]
Anticonvulsant	Carbamazepine	Blocking of central Na^+^ channel	CHOL, LDL	Yes [85]	Yes [86,87,88]	Yes [86,87,88]
	Tiagabine	GABA reuptake inhibition	CHOL			
Antidepressant	Paroxetine	Selective serotonin reuptake inhibition	CHOL			
	Sertraline	Selective serotonin reuptake inhibition	CHOL			
	Duloxetine	Serotonin and noradrenaline reuptake inhibition	CHOL			
	Venlafaxin	Serotonin and noradrenaline reuptake inhibition	CHOL			
Antigonadotropic	Danazol	Androgen receptor activation	CHOL, LDL			
Antigout	Febuxostat	Xanthine-oxidase inhibition	CHOL			
Antihyperglycemic	Ertugliflozin	SGLT-2 inhibition	CHOL, LDL			
	Sotagliflozin	SGLT1/2 inhibition	CHOL, LDL			
Antihypertensive	Lacidipine	Ca^2+^ channel blocker	LDL	Yes [89]		
	Furosemide	Diuretic	CHOL, LDL			
	Indapamide	Diuretic	CHOL, LDL			
	Propranolol	Beta-blocker	CHOL			
Antimycotic	Fluconazole	Ergosterol synthesis inhibition	CHOL			
	Voriconazole	Ergosterol synthesis inhibition	CHOL			
Antineoplastic	Apalutamide	Antiandrogen	CHOL, LDL	Possible [90]	Yes [90]	Yes [90]
	Anastrozole	Aromatase inhibition	CHOL			
	Letrozole	Aromatase inhibition	CHOL			
	Mitotane	Adrenal cortex inhibition	CHOL, LDL	Yes [32]	Yes [32,91]	Yes [92]
	Asparaginase	Depletion of circulating asparagine	CHOL			
	Histrelin	GnRH agonist	CHOL			
	Degarelix	GnRH blocker	CHOL			
	Pegvisomant	IGF1 inhibition	CHOL			
	Ruxolitinib	JAK inhibition	CHOL	Possible ****		
	Rucaparib	PARP inhibition	CHOL			
	Verteporfin	Phototherapy sensitizer	CHOL			
	Cladribine	Purine analogue	CHOL			
	Tegafur	Pyrimidine analogue	CHOL, LDL			
	Padeliporfin	Radiation therapy sensitizer	CHOL, LDL			
	Brigatinib	Tyrosine kinase inhibition	CHOL, LDL	Possible *	Yes *	
	Cabozantinib	Tyrosine kinase inhibition	CHOL, LDL			
	Dasatinib	Tyrosine kinase inhibition	CHOL, LDL	Yes [89]		
	Lenvatinib	Tyrosine kinase inhibition	CHOL, LDL		Yes	
	Lorlatinib	Tyrosine kinase inhibition	CHOL, LDL	Yes **	Yes **,***	
	Nilotinib	Tyrosine kinase inhibition	CHOL, LDL	Yes [93]		
	Pazopanib	Tyrosine kinase inhibition	CHOL			
Antipsychotic, atypical	Amisulpride	Inhibition of D2 and 5-HT2A receptors	CHOL			
	Aripiprazole	Inhibition of D2 and 5-HT2A receptors	LDL			
	Cariprazine	Inhibition of D2 and 5-HT2A receptors	CHOL, LDL			
	Clozapine	Inhibition of D2 and 5-HT2A receptors	CHOL, LDL		Yes [94]	
	Olanzapine	Inhibition of D2 and 5-HT2A receptors	CHOL, LDL			
	Paliperidone	Inhibition of D2 and 5-HT2A receptors	CHOL, LDL			
	Quetiapine	Inhibition of D2 and 5-HT2A receptors	CHOL, LDL	Yes [95]	Yes [95]	
	Risperidone	Inhibition of D2 and 5-HT2A receptors	CHOL, LDL			
Antipsychotic, typical	Fluphenazine	Inhibition of D2 receptors	CHOL			
	Zuclopenthixol	Inhibition of D2 receptors	CHOL, LDL			
Antiretroviral	Cobicistat	CYP3A inhibition	CHOL			
	Raltegravir	Integrase inhibition	CHOL, LDL			
	Efavirenz	Non-nucleoside reverse transcriptase inhibition	CHOL, LDL	Yes [50]	Yes [50,96,97]	Yes [97,98]
	Etravirine	Non-nucleoside reverse transcriptase inhibition	CHOL, LDL	Yes [99]	Yes [100,101,102]	Yes [100,101]
	Nevirapine	Non-nucleoside reverse transcriptase inhibition	CHOL, LDL		Yes [103,104,105]	Yes [103,104,105]
	Rilpivirine	Non-nucleoside reverse transcriptase inhibition	CHOL, LDL	Yes [99]		
	Darunavir	Protease inhibition	CHOL	Yes [50]		
	Fosamprenavir	Protease inhibition	CHOL	Yes [106]	Yes [106]	Yes [106]
	Indinavir	Protease inhibition	CHOL			
	Lopinavir	Protease inhibition	CHOL	Yes [50]		
	Ritonavir	Protease inhibition	CHOL	Yes [107]	Yes [108,109]	Yes [108,109]
	Saquinavir	Protease inhibition	CHOL, LDL	Yes [107]		
	Tipranavir	Protease inhibition	CHOL		Yes *****	Yes *****
Antithyroid	Methimazole	Thyroperoxidase inhibition	LDL			
Antiviral	Boceprevir	Protease inhibition	CHOL			
Barbiturate	Phenobarbital	GABA stimulation	LDL	Yes [110]	Yes [111]	Yes [111]
Emergency contraception	Ulipristal	Progesterone receptor modulation	CHOL			
Immunosuppressant	Cyclosporin	Calcineurin inhibition	CHOL, LDL	Yes [112]		
	Tacrolimus	Calcineurin inhibition	CHOL	Yes [113]		
	Rituximab	CD20 inhibition	CHOL, LDL			
	Beclomethasone	Glucocorticoid receptor activation	CHOL	Yes [114]		
	Dexamethasone	Glucocorticoid receptor activation	CHOL	Yes [115]	Yes [116,117,118,119]	Yes [117,118,119]
	Prednisolone	Glucocorticoid receptor activation	CHOL		Yes [120,121]	Yes [120,121]
	Prednisone	Glucocorticoid receptor activation	CHOL		Yes [116,122]	Yes [122]
	Anakinra	IL-1 inhibition	CHOL			
	Rilonacept	IL-1 inhibition	CHOL, LDL			
	Basiliximab	IL-2 inhibition	CHOL, LDL			
	Sarilumab	IL-6 inhibition	CHOL, LDL			
	Siltuximab	IL-6 inhibition	CHOL			
	Tocilizumab	IL-6 inhibition	CHOL, LDL			
	Baricitinib	JAK inhibition	CHOL, LDL			
	Tofacitinib	JAK inhibition	CHOL, LDL			
	Everolimus	mTOR inhibition	CHOL, LDL			
	Sirolimus	mTOR inhibition	CHOL			
	Temsirolimus	mTOR inhibition	CHOL, LDL			
	Leflunomide	Pyrimidine synthesis inhibition	CHOL, LDL			
	Mycophenolate mofetil	Purine synthesis inhibition	CHOL			
	Adalimumab	TNF inhibition	CHOL, LDL			
	Certolizumab Pegol	TNF inhibition	CHOL, LDL			
	Golimumab	TNF inhibition	CHOL, LDL			
	Infliximab	TNF inhibition	CHOL, LDL			
Non-steroidal anti-inflammatory drug	Acetylsalisylic acid	COX inhibition	CHOL		Yes [123]	
	Diclofenac	COX inhibition	CHOL			
	Ibuprofen	COX inhibition	CHOL			
Other	Ataluren	Ribosome function modulation	CHOL			
	Human normal immunoglobulin	Improved pathogen removal	CHOL			
	Leuprorelin	GnRH analogue	CHOL, LDL			
Proton pump inhibitor	Lansoprazole	Stomach acid reduction	CHOL	Yes [124]	Yes [124]	
	Pantoprazole	Stomach acid reduction	CHOL			
Antioxidant	Idebenone	Mitochondrial electron transport chain stimulation	CHOL			
Retinoid	Alitretinoin	Retinoid receptor activation	CHOL		Yes [125]	
	Bexarotene	Retinoid receptor activation	CHOL, LDL		Yes [126,127]	Yes [126,127]
	Isotretinoin	Retinoid receptor activation	CHOL		Yes [125]	
	Tretinoin	Retinoid receptor activation	CHOL		Yes [125]	
Stimulant	Modafinil	Dopaminergic modulation	CHOL	Yes [128]	Yes [129]	
Vitamin	Cholecalciferol	Vitamin D receptor activation	CHOL, LDL			

* U.S. Food and Drug Administration Center for Drug Evaluation and Research (2016) Chemistry review on Brigatinib (application number 208772Orig1s000), available at https://www.accessdata.fda.gov/drugsatfda_docs/nda/2017/208772Orig1s000ChemR.pdf (accessed on 13 December 2021). ** Pfizer (2021) Highlights of LORBRENA (lorlatinib) prescribing information, available at https://www.accessdata.fda.gov/drugsatfda_docs/label/2021/210868s004lbl.pdf (accessed on 13 December 2021). *** European Medicines Agency (2021) Summary of Lorviqua (lorlatinib) product characteristics, available at https://www.ema.europa.eu/en/documents/product-information/lorviqua-epar-product-information_en.pdf (accessed on 13 December 2021). **** U.S. Food and Drug Administration Center for Drug Evaluation and Research (2011) Genomics group review on ruxolitinib (application number 202192Orig1s000), available at https://www.accessdata.fda.gov/drugsatfda_docs/nda/2011/202192Orig1s000ClinPharmR.pdf (accessed on 13 December 2021). ***** U.S. Food and Drug Administration Center for Drug Evaluation and Research (2004) Pharmacology/toxicology review and evaluation (application number NDA 21-814), available at https://www.accessdata.fda.gov/drugsatfda_docs/nda/2005/21814_000_Aptivus_pharmr1.pdf (accessed on 13 December 2021).

## Data Availability

The data presented in this study are available within this article.

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
