# Peer review of "Nuclear Receptor PXR in Drug-Induced Hypercholesterolemia"

_cells, 2022, doi:10.3390/cells11030313_

Round 1
Reviewer 1 Report
In this manuscript entitled ‘Nuclear receptor PXR in drug-induced hypercholesterolemia’, Karpale et al provided a comprehensive review on the role of pregnane X receptor (PXR), a key nuclear receptor mediating drug and xenobiotic metabolism, in drug-induced hypercholesterolemia. The authors summarized the evidence of PXR-induced hypercholesterolemia in human clinical studies and discussed the potential mechanisms of PXR-induced hypercholesterolemia. The authors also provided a comprehensive table including more than 100 clinically used drugs that increase total cholesterol or LDL-cholesterol and discussed those were identified to be PXR ligands. Overall, the manuscript is well organized, the written is clear, and the discussion is appropriate. I have the following specific comments:
- The abstract and introduction started with background information about atherosclerosis, which closely associated with hypercholesterolemia. However, authors then mentioned that hypercholesterolemia is also involved in the pathogenesis of many other diseases, and the focus of the present manuscript is ‘hypercholesterolemia’ instead of ‘atherosclerosis’. It would be appropriate if the authors reorganize these parts and start the paper with hypercholesterolemia instead of atherosclerosis.
- Section 4 (PXR in HDL homeostasis) is relatively short and could be a part of section 3 instead of a separated section.
- Figure 1 could be improved. It’s not clear how PXR regulates these pathways. It would be better if the authors could modify this one or create a new picture to elucidate the different mechanisms for PXR-induced hypercholesterolemia.
- There are several typos, grammar errors and duplicated references. More careful proofreading should be conducted during revision. Examples: Line 62: “‘individuals drugs”; References 100 and 160 are the same paper, etc.
Author Response
- The abstract and introduction started with background information about atherosclerosis, which closely associated with hypercholesterolemia. However, authors then mentioned that hypercholesterolemia is also involved in the pathogenesis of many other diseases, and the focus of the present manuscript is ‘hypercholesterolemia’ instead of ‘atherosclerosis’. It would be appropriate if the authors reorganize these parts and start the paper with hypercholesterolemia instead of atherosclerosis.
Response: We have modified the introduction as suggested by the reviewer and start with the definition of hypercholesterolemia. However, since the atherosclerosis is the most important disease related to hypercholesterolemia and for the flow of the text, we have left the abstract in its original order.
- Section 4 (PXR in HDL homeostasis) is relatively short and could be a part of section 3 instead of a separated section.
Response: We understand the reviewer’s point. However, the section 4 shortly discusses the potential beneficial effects launched by PXR activation and therefore it does not fit very well under the subtitle “Mechanisms of PXR-induced hypercholesterolemia”. Therefore, we have kept section 4 as it is.
- Figure 1 could be improved. It’s not clear how PXR regulates these pathways. It would be better if the authors could modify this one or create a new picture to elucidate the different mechanisms for PXR-induced hypercholesterolemia.
Response: We have made some modifications to the figure 1 to make it more illustrative. Furthermore, we have modified the figure legend to better convey the message of the figure.
- There are several typos, grammar errors and duplicated references. More careful proofreading should be conducted during revision. Examples: Line 62: “‘individuals drugs”; References 100 and 160 are the same paper, etc.
Response: We have proofread the manuscript and corrected several typos and errors we could notice. Furthermore, the duplicate references have been deleted.
Reviewer 2 Report
Major Comments:
- I found the introduction difficult to follow, it jumps around and repeats itself. I’d suggest some editing there. For example, paragraph in line 113-116 is two sentences and basically repeats their comments regarding lathosterol and cholesterol synthesis.
- The review of the changes in cholesterol with PXR inducing drugs is too detailed; however, then in line 91 the length of the studies is not discussed which is important for determining if the cholesterol changes are transient.
Minor Comments
- In line 132, I’d suggest that point 2 – ‘may’ play a role rather than such a definitive remark.
- They need to at least acknowledge the differences in PXR activation between humans/rats vs. mice.
- The section on PCSK9 makes sense, but is a string of short, 2-sentence paragraphs which chops it all up. The authors could consider the flow of the language from section to section to make the review more digestible to the readers.
- Section 5. could be substantially shortened, as it drifts away from the intention of the review manuscript.
Author Response
Major Comments:
- I found the introduction difficult to follow, it jumps around and repeats itself. I’d suggest some editing there. For example, paragraph in line 113-116 is two sentences and basically repeats their comments regarding lathosterol and cholesterol synthesis.
Response: We have done some editing of the introduction section to improve it. However, we were not sure if the reviewer was actually referring to the section 2 including lines 113-116. We have made some corrections also to the section 2 to improve readability and have also edited the sentences in lines 116-117 (previously 113-116).
- The review of the changes in cholesterol with PXR inducing drugs is too detailed; however, then in line 91 the length of the studies is not discussed which is important for determining if the cholesterol changes are transient.
Response: This comment refers to section 2 that was partly discussed already in the response to the previous comment. Concerning the too many details, we have improved readability of the text by removing decimals from the numbers. We believe that it is important and useful for the reader to collect the current clinical evidence pointing towards a role of PXR in cholesterol elevating effect of drugs. We agree with the reviewer that the length of studies is important information and have added this missing piece of data (line 94).
Minor Comments
- In line 132, I’d suggest that point 2 – ‘may’ play a role rather than such a definitive remark.
Response: This has been corrected as suggested by the reviewer. line 135
- They need to at least acknowledge the differences in PXR activation between humans/rats vs. mice.
Response: We have added a note on this, lines 149-151.
- The section on PCSK9 makes sense, but is a string of short, 2-sentence paragraphs which chops it all up. The authors could consider the flow of the language from section to section to make the review more digestible to the readers.
Response. We have combined the two short paragraphs in the section 3.2. discussing PCSK9 results. In general, we do not agree with the reviewer that the long paragraphs would increase the readability of the text.
- Section 5. could be substantially shortened, as it drifts away from the intention of the review manuscript.
Response: We tend to think that this section is an important part of this review as it describes the magnitude of problem and may attract also more clinically oriented readers for the review. Furthermore, the section 5 forms the basis for the section 6, which then discusses the potential overall significance of PXR activation on drug-induced hypercholesterolemia. However, we have made an effort to concise the section and have removed some unessential information (several parts in section 5).